# Exploring the Potential of Digital Game-Based Vocabulary Learning: A Systematic Review

**Gregor Vnucko** [1,*] and **Blanka Klimova** [2]

1. Department of English Language and Culture, Faculty of Pedagogy, Constantine the Philosopher University in Nitra, Drážovská 4, 949 01 Nitra, Slovakia
2. Department of Applied Linguistics, Faculty of Informatics and Management, University of Hradec Kralové, Hradecká 1249/6, 500 03 Hradec Králové, Czech Republic
* Correspondence: gregor.vnucko@ukf.sk

**Abstract:** Constant technological development creates the need for continual optimization of the educational process to achieve educational objectives with the highest possible efficiency. Surveys have shown that digital games are very popular among all age groups, including when they are used in foreign language education, such as in Digital Game-Based Vocabulary Learning (DGBVL). Therefore, the purpose of this review study is to summarize the existing knowledge on DGBVL and to provide new, cutting-edge information based on available literature accessible within three databases: Web of Science, Scopus, and ScienceDirect. After applying the inclusion and exclusion criteria and conducting the search, 13 articles were identified for a deeper analysis. The main findings reveal that DGBVL can create a positive learning environment. In this kind of environment, students experience predominantly positive emotions which may in turn enhance their vocabulary retention. The review suggests that DGBVL is useful in vocabulary learning and may even surpass conventional teaching methods in an English classroom. The findings of this review can serve as a basis for further research, which could ultimately lead to the implementation of DGBVL in the process of English instruction, not only as a supplementary method, but possibly as a full alternative to conventional English lessons.

**Keywords:** digital games; vocabulary learning; DGBVL; playing games; mental lexicon

## 1. Introduction

Technology has affected all areas of our life, from our simple daily routines (brushing our teeth with an electric toothbrush), to how we spend our free time (watching TV, playing video games) or achieve our professional objectives. Statistical data collected by the Interactive Software Federation of Europe (ISFE, 2021) [1] showed that, among citizens of five major European countries (France, Italy, Germany, Spain, United Kingdom), every other person aged 6 to 64 years old played video games. In Figure 1 below, the share of video game players according to individual age groups is illustrated. The data revealed that digital games were popular among all age groups. The share of players was as high as 84% among child survey participants (11–14 y.o.). However, even within the oldest survey group (45–64 y.o.), almost every third person claimed to have played video games [1].

Even though the outbreak of the COVID-19 pandemic suggested social distancing and isolation, the results show that this did not lead to an increase in the share of video game players (as can be observed in Figure 1 below) [1]. The average player's gaming time was reported to be 9.5 h per week and 47% of all players participating in the survey reported being female.

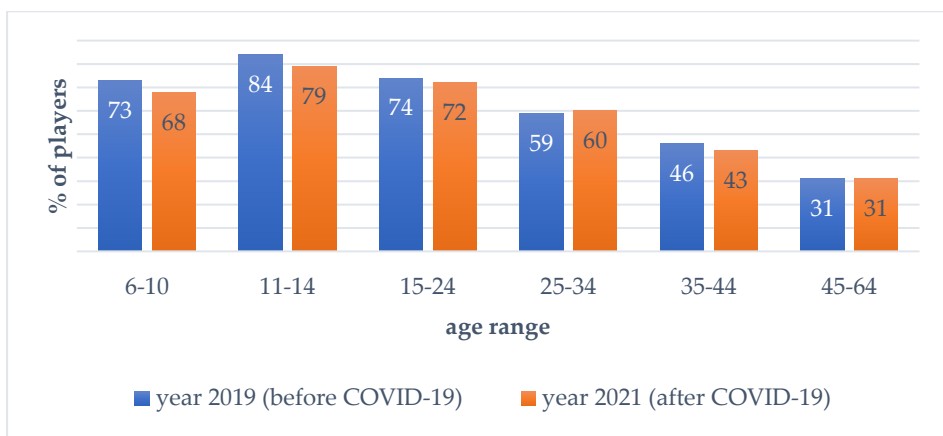

**Figure 1.** Share of video game players by age group according to ISFE [1].

A larger survey was conducted by the Statistical Office of the European Union (2019) [2] and included most European countries. The data from this survey (collected in 2018) reported that 33% of all people aged 16–74 had played video games. This number is even higher overseas. ESA [3] suggests that 66% of all Americans play video games, and 97% of all Americans believe that playing digital games can be beneficial in some way.

However, many of the digital games that teenagers play have not yet been translated into their native language. Therefore, to participate in the game and achieve its objectives, they need to be exposed to the English language and use it actively. These digital games have also affected their learning, a phenomenon which has become known as digital game-based learning (hereinafter DGBL). DGBL is defined as learning through applications that were designed to meet educational goals and to achieve desired educational outcomes [4,5]. Additionally, DGBL allows students to explore different parts of the game while at the same time develop their skills and expand their knowledge [6]. The concept of game-based learning has been around for centuries, but utilizing video games in language learning has become popular mainly since the 2010s [7].

The initial research on DGBL started at the end of the 20th century [8,9], and the first significant breakthrough with respect to connecting the use of digital games and learning was described by Gee (2003) [10] and Prensky (2001) [11]. Their efforts helped to open a new approach to digital gaming and recognized it as a potential tool for learning, thus establishing DGBL.

When dealing with digital games in language learning, teachers and researchers should consider some specific traits characteristic of different age groups. Within childhood education, play is suggested to be an effective tool, stimulating cognition and learning while at the same time promoting healthy brain development [12–14]. Therefore, play should be an integral part of any digital game which is being considered for the use in instruction. Additionally, parents demonstrated positive perceptions and attitudes towards the incorporation of digital games in children's early school instruction [15]. To optimize the teaching/learning process of DGBL, educational games utilized in childhood education should include a straight-forward design, narrative, simple game mechanics, and immediate feedback. Other features which may prove valuable are rewards, content, graphic sounds, and the overall user experience. Examples of such games include "Decimal Point" [16], "The MOBO City" [17], and "Trash Attack" [18]. These key features of a "good" educational game where later modified and improved by Garcia (2020) [19], to create a digital visual novel game called "Kinder Learns". This game proved to be a successful tool when used in various school subjects, such as mathematics, science, art, physical education, and language learning.

Instructional digital games provide adults with more motivation when the goals are clearly stated and when they do not feel overwhelmed by the learning process. Additionally, control over game progression may in turn increase their activity and engagement by

offering a personalized experience. This kind of environment might deliver feelings of safety and protection, where adult learners feel less anxious about making and correcting their mistakes [20]. On the other hand, unlike children, it might not be possible to label adults as "digital natives" [21], as they were not raised with the today's technology. Because some adults (especially the elderly) might not be very well acquainted with different digital media, it is advised to offer constant technological support [20]. When designing an educational game for adults, it is recommended to activate all sensory inputs and include as much variety as possible, as the strengths of individual students may vary [20].

Digital games for language learning can facilitate the development of all four major language skills, and can additionally develop their vocabulary and enrich grammatical competence [7,22]. Research investigating the learning outcomes of digital game-based learning has reported on several areas. These include the cognitive area, such as learning outcomes; behavioural outcomes; and either affective outcomes [23], motivational outcomes [24], or both [25]. Motivational outcomes are often characterized as a subcategory of affective outcomes [26], similar to Bloom et al. (1984)'s [27] taxonomy of educational objectives [28]. Digital games can increase exposure to the target language, and have also been reported to increase engagement in learners [20,22].

Apart from digital game-based learning, Reinhardt and Sykes (2012) [29] also distinguished another very closely connected term: game-enhanced learning. Game-enhanced learning refers to the use of commercial off-the-shelf digital games (hereinafter COTS games), which were then utilized for learning. These games include titles enjoying popularity among society at large. In game-enhanced learning, the researchers do not participate in the design of the game but tailor the educational activities and techniques according to the game. Game-enhanced learning is learning itself aided by the COTS games [5]. As opposed to DGBL, which uses "serious games", COTS games are genuine, designed for the broader society, and often include different cultural narratives which may also increase cultural awareness, apart from language learning. These sorts of games, such as the genre of massively multiplayer online (MMO) games, are often enjoyed by people from all over the world, and thus may include a lot of engagement with native speakers [30]. In addition, they are also popular in foreign language learning, especially in learning new words and phrases.

## 2. Literature Review

Many researchers, scholars, students, and teachers agree that vocabulary is a substantial part of mastering any language. To integrate words into the learner's mental lexicon, it is important to transform their initial explicit (receptive, conscious, passive, declarative) knowledge into implicit (productive, subconscious, active, procedural) knowledge to be able to use the vocabulary in authentic communication [31]. The available research has shown that using digital games in the context of developing vocabulary has proven effective and has great potential [30,32–45].

Rasti and Vahdat (2013) [41] revealed within their study that gender differences may also play a significant role in the use of Digital Game-Based Vocabulary Learning (hereinafter DGBVL), by finding a positive correlation between gender and vocabulary learning through video games. By contrast, Saprudin et al. (2019) [46] found that the increase in academic achievement by students in large classes was notably higher than the achievement of students in small classes, which could suggest that with the use of DGBVL, class size may no longer be a barrier to implementing active learning in the future. To answer the question "how can digital games be advantageous in vocabulary learning?", Rasti (2021) [47] observed seven reoccurring factors that digital games provide (p. 112): *high level of motivation, different types of repetition, various types of feedback, authentic contexts for learning in their virtual worlds, a rich context for dual encoding, high degree of interactivity, manipulation and control over the content, rich instantiations of words*.

Based on the currently available literature, there is no doubt that DGBVL can bring many benefits to its users. However, it is also necessary to point out the challenges or

problems that may arise while using this approach. Mayer (2014) [48] offered such an example by stating that many times serious games did not surpass traditional learning with respect to learning outcomes. According to Chen et al. (2016) [35], the next variable in determining the effectiveness of DGBVL is the game's genre. Adventure game genres can be more stimulating and interesting, and can also motivate the students more, as they require more complex cognitive processes, such as critical thinking, problem-solving, or task engagement, as opposed to non-adventure game genres. Furthermore, being immersed in the game itself does not always guarantee that the gaming participant is immersed in the target learning topic. Additionally, another study [30] pointed out that this kind of immersion may bring additional cognitive load onto the students, which may hinder their learning process, including vocabulary retention. This is in concordance with Jonathan et al. (2010)'s [39] study, which suggested that games put an additional cognitive load on their players, who may forget many newly acquired vocabulary items. Emotions are an inseparable part of play and represent a crucial role in any form of learning [49], and have a significant impact on vocabulary retention [50].

Serious games have the capacity to enhance the student's cognitive abilities and at the same time minimize the negative effects resulting from making errors [51]. In this kind of virtual environment, the learners can explore different paths and solutions without being nervous about facing the real consequences of those actions, which allows them to manage errors in a more positive way [52]. This opportunity may reduce the anxiety the students face when making mistakes within regular classrooms (e.g., erring in front of their peers), as the virtual environment may allow the students to make mistakes freely in an enjoyable and protected environment (ibid.). To ensure that the learners treat errors as an important educational factor, positive feedback is incorporated within the games to encourage the individuals to set or accept higher goals [53,54]. Serious games, in which making errors is seen as a natural and foreseen part of the educational experience, have higher feasibility of creating a learning experience with predominantly positive emotions [49]. Activities resulting in positive emotions for the learners, in turn, enhance their memory, enabling students to retain more information and to recall it in more detail (ibid.). Among the previously mentioned affective factors, while playing a game, learners may also experience feelings of competition, engagement, satisfaction, or fulfilment after completing the achievements, all while moving towards developing the desired educational outcomes [7].

As has been mentioned above, technological development is moving fast, and technology is becoming part of our daily lives. Scientists have seen the opportunity that digital games bring and have focused on seizing this opportunity for several decades. DGBVL is a vast subject that exposes the need for further exploration. Innovations within the educational process are most welcome if they possess the ability to increase the effectiveness of that process. An integral part of language learning is vocabulary learning and there is a need for an analysis of DGBVL, which would offer scientists who want to delve into this topic in the future a synthesized resource and a clear and organized review of the recent DGBVL literature.

Thus, the main aim of this paper is to explore the existing and cutting-edge knowledge on DGBVL, based on available literature accessible within three databases: Web of Science, Scopus, and ScienceDirect. To accomplish this goal, the following research questions have been set:

- **RQ1:** What are the main benefits and limitations of DGBVL?
- **RQ2:** What are the common trends researchers follow to conduct research in the area of DGBVL?
- **RQ3:** Is DGBVL useful in learning English as a foreign language?
- **RQ4:** What are the pedagogical implications stemming from the available DGBVL research within the last 5 years?

## 3. Method

*Data Sources and Search*

For collecting the research data, the authors of this review followed the Preferred Reporting Items for Systematic Reviews and Meta-Analyses [55], which has provided a reliable structure for the method of performing a systematic review. The PRISMA statement should provide a transparent account of the review.

Due to their availability, three databases were used for conducting the review: Web of Science (WoS), Scopus, and ScienceDirect. Journal entries from the period of January 2018 to July 2022 were searched. Articles had to be in English, from a peer-reviewed journal. The authors of this review searched for combinations of the following keywords: gam *, play *, language learning, vocabulary, and lexis. Because the search function in ScienceDirect allowed only 8 Boolean operators, the search string was simplified as follows: ("Play" AND "language learning" AND "vocabulary") OR ("Game" AND "language learning" AND "vocabulary") OR ("Game" AND "language learning" AND "lexicon"). The exclusion and inclusion criteria set for the search were as follows:

Inclusion criteria

1.  Articles containing a combination of selected keywords within their abstract, title, or keywords.
2.  Peer-reviewed journal articles.
3.  Only in the English language.
4.  Journal entries only within the period from January 2018 to July 2022 (i.e., the last 5 years).
5.  Full text available/open access.
6.  Only research papers with empirical research.

Exclusion criteria

1.  Publications not available as open access.
2.  Document types other than journal articles.
3.  Articles in the press.
4.  Publications in a language different than English.
5.  Journal entries older than January 2018.
6.  Review articles.
7.  Publications concerned with traditional non-digital classroom games.

The first search in Scopus initially showed 425 results, which were reduced to 57 after applying the exclusion criteria. In WoS, initially, 468 results were found, while after limiting the search using exclusion criteria only 61 remained. For ScienceDirect, 33 results were found, and after applying exclusion criteria, only 10 remained.

In summary, 926 results were initially found, while 128 remained for further inspection. After removing duplicates, this number was reduced to 94 articles. Based on the titles and abstracts, 27 articles were left for full-text analyses. Furthermore, 14 more full-text records were removed as they did not correspond to the purposes of this research. Thirteen full-text records remained, which were also included in this systematic review (Figure 2).

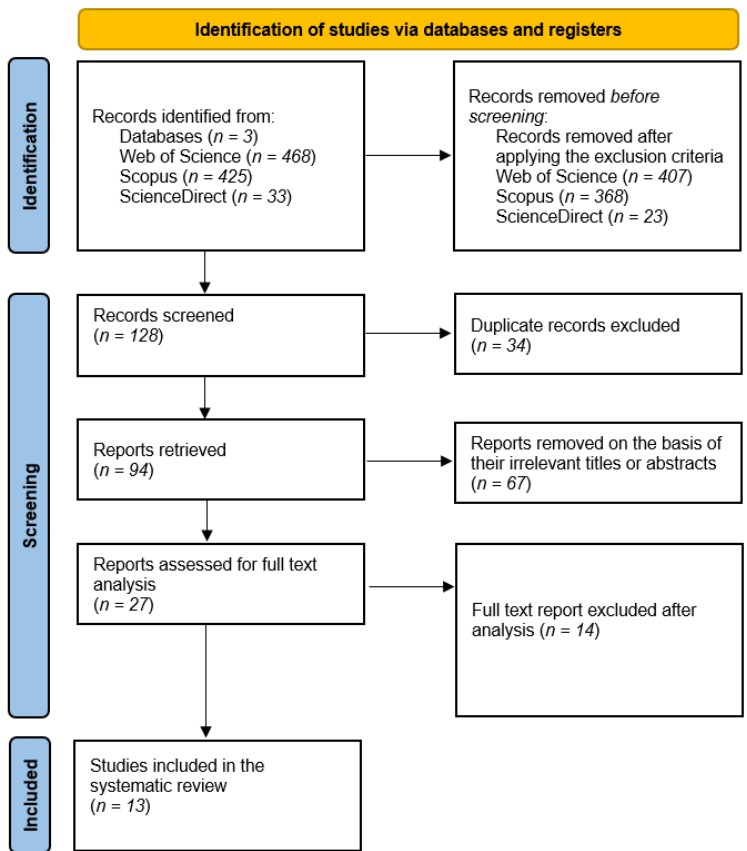

**Figure 2.** PRISMA flow diagram.

### 4. Results

Altogether 13 studies were detected. Out of thirteen available records, six papers followed a quasi-experimental design with experimental and control groups, from which all six found that learning vocabulary through the use of digital games is effective. Additionally, four of them found that the experimental group (EG) outperformed the control group (CG) in both immediate and delayed post-tests. The sample subjected to the quasi-experiments were people from various age groups, ranging from higher grades of elementary school (11 yr.), up to seniority (60 yr.). In addition, teenagers formed the most frequent research group. Within the category of methods and instruments, the most commonly used methods were quasi-experiments, semi-structured interviews, and questionnaires. The structure of the semi-structured interviews, as well as the questionnaires, was always designed or modified by the researcher. There were no standardized tests or structures used within the analysed journal entries, apart from the preliminary screening in two papers, which used Oxford placement tests to ensure that the students are of the same level of language proficiency. It could be thus hypothesized that there are no standardized tests yet available for testing gaming vocabulary. This could be due to the variable nature of games.

Within the available and accessible empirical research from the last 5 years, the authors of the detected articles have achieved different positive outcomes. Digital games in language learning have been reported to create an engaging learning environment [56–59], and to provide an interesting way of learning [56]. The gaming participants may learn new vocabulary in a risk-free environment [59]. With the combination of a variety of contextual vocabulary [57], vocabulary learning via digital games may become easier for the students [60], build extra self-confidence, promote autonomous learning behaviours, or increase motivation [57,61]. The most common vocabulary learning strategies utilized within autonomous learning were consultation, incorporating words with a real word, and using an online dictionary [62]. Additionally, based on interviews, many students stated

that they would be happy if they could learn vocabulary via the use of digital games, and that they had overall positive attitudes towards this approach [60,63,64].

On the other hand, there are also some challenges that need to be addressed. Using games for the sake of entertainment without employing a learning structure has proven less effective than using intentional learning strategies whilst gaming [65]. Thus, it is recommended to always employ intentional learning or teaching strategies, both to facilitate incidental vocabulary acquisition and to make learning new vocabulary even more effective [66]. Additionally, the teachers employing games for vocabulary learning need to keep in mind that this method may be more efficient in developing productive recognition of form–meaning relationships, in contrast to traditional teaching methods, which proved to develop more receptive knowledge of acquisition [67]. The teachers need to be careful when choosing the right game type, and the game itself, because as both Deris and Shukor, (2019) [64] and Fu et al., (2021) [61] suggest, research participants often found vocabulary within the games too complex and challenging, which may pose a hindrance in utilizing this method of incorporating games and vocabulary learning. Furthermore, many video games, especially mobile serious games, may require an internet connection to be fully used, which proved problematic for many research participants [64].

As can be seen in Table 1 Below, in the first column, the countries of the authors and journal articles can indicate which countries are the most interested in the issue of DG-BVL. Only five out of thirteen studies available were conducted in Europe [58,59,63,65,67]. Most of the studies were conducted in Asia, namely in Indonesia [56,66,68], India [57], Bangladesh, Saudi Arabia [60], Malaysia [62,64], and China [61]. European authors contributing to the topic of DGBVL came from Finland [67], France [63], and Spain [58,65]. One study was conducted at the National Defence University in Istanbul [59]; thus, it is classified as both a European and Asian journal entry. The data have shown that within the period of the last 5 years, no articles were produced within central Europe, even though DGBVL has shown effectiveness in vocabulary learning. Conducting future studies in countries in central Europe could potentially offer a different perspective on the data regarding the topic of DGBVL.

The next category which can be observed from the results within Table 1 includes the limitations of the studies. The limitations were mainly identified with respect to the sample, time constraints, the extent of the studies, and the tools used within the research. Some studies suggested that they may not have included a sufficient number of participants to draw broad generalizations based on those studies [57,59,62,64]. Furthermore, the focus of those studies was often limited to only one age group, such as tertiary students [61,64,66]. Rahman and Angraeni (2020) [68] warned that both the language proficiency of the sample and the digital game used in the teaching/learning process must be carefully taken into consideration, as sometimes the text coverage within the digital game may prove too challenging. Within the sample, most of the studies which disclosed the sample's gender had a predominantly male sample, which could affect the study results [41]. However, a more recent study [65] has shown that gender might not be a variable in affecting the results of DGBVL instruction. Other authors stated that they may have had insufficient time allocated for their research [68]. This disallowed them from tailoring the game design to its users' needs [56,58], or from completing a follow-up vocabulary test with the participants to check whether the participants retained vocabulary items in their long-term memory [61]. Attention also needs to be allocated to the choice of the learning media, as the studies often focused on only one digital medium, such as the personal computer [65], or iPad [59]. Additionally, the pre-tests completed with the participants may have helped them achieve higher scores in the post-test, due to the repetitive nature of those tests, even though the sequencing was administered carefully [65,67].



**Table 1.** Overview of key findings.

| Authors, Year, Country | Research Aim(s)/Research Question(s) | Sample Age No. of Participants Control/Experimental Groups–Yes/No? | Types of the Game(s) | Methods/Instruments | Major Findings | Pedagogical Implications | Limitations |
|---|---|---|---|---|---|---|---|
| **Andreani and Ying, 2019 [56], Indonesia** | Propose a learning alternative approach using an interactive game. | 7–12 yr. 112 in design, 35 in play No | A serious game designed by the researchers | Opinion pre- and post-questionnaires | The game provides an interesting way of learning. The game builds an engaging learning environment. | The serious game has succeeded in attracting students' interest in learning English and in increasing their motivation. Additionally, after further research, it could be used for vocabulary learning in the future. | More practice time would be required for developing the application tailored to users' needs. |
| **Peake and Reynolds, 2020 [63], France** | Examine student attitudes to language learning in relation to online video gaming in English as a leisure activity. | 21 yr. average 90 No | Commercial-off-the-shelf games | Interviews, questionnaires, photographs | The game helped participants to learn new vocabulary without the impression of being drilled. Gaming may create positive attitudes toward learning languages. | The results showed that student participants had no negative associations with gaming in English, whereas their attitudes to formal language learning were more mixed. Implementation of such games in FLT could increase motivation and reduce anxiety in students. | Not included. |
| **Patra et al., 2022 [57], India, Saudi Arabia and Bangladesh** | Evaluate the impact of playing digital games on the ability of Bangladeshi EFL learners' to remember and retain the language. | 13–17 yr. 50 Yes | Serious game | Quasi-experimental study (pre-test, treatment, and post-test), Oxford placement test | EG outperformed CG in vocabulary post-test. Learning via games offers more engagement than traditional learning methods. Games are a source of motivation. Games build confidence in students. Games allow a variety of contexts for the lexical items. | This study can assist teachers in improving their teaching quality by using games. The variety of games used in FLT could simplify the process of clarification for students. | Insufficient number of participants for a representative sample of a big population. Gender—all the participants were male college students. Proficiency level—only A1. This study uses only quantitative data. |

**Table 1.** *Cont.*

| Authors, Year, Country | Research Aim(s)/Research Question(s) | Sample Age No. of Participants Control/Experimental Groups–Yes/No? | Types of the Game(s) | Methods/Instruments | Major Findings | Pedagogical Implications | Limitations |
|---|---|---|---|---|---|---|---|
| **Abdullah, 2020 [60], Saudi Arabia** | 1. What were students' preferences regarding the use of Quizlet in English vocabulary learning? 2. How often did students use Quizlet in English vocabulary learning outside the classroom? 3. What were students' attitudes towards the use of Quizlet in English vocabulary learning? | 18–22 yr. 38 No | Serious mobile game | Questionnaires, semi-structured interviews | Participants showed a positive attitude towards learning via a mobile app. Participants viewed the serious game as a tool that made vocabulary learning easier. | The serious mobile game is perceived as a useful tool that can be utilized inside and outside the classroom. It is recommended to select user-friendly tools (games) that offer a range of learning features that can meet the different goals and needs of learners. | Not included. |
| **Rahman and Angraeni, 2020 [68], Indonesia** | Investigate the significant effect of an RPG on students' vocabulary mastery and also their responses toward the media. | High school students (age not disclosed) 65 Yes | Adventure commercial game | Quasi-experimental design (pre-test, treatment, and post-test), interviews, questionnaires | EG outperformed CG in vocabulary post-test. Participants showed a positive attitude towards learning via the game. The game proved to be effective as a medium for learning new vocabulary. | Teachers also found this media helpful because the media can easily be applied and is effective, particularly to motivate learners and to retain the vocabulary terms for a longer period. | Proficiency level—participants had difficulty in understanding the text coverage. DGBVL could be used to develop other language skills and systems. Lack of time may have brought other factors which could have affected the results of the study. |

**Table 1.** *Cont.*

| Authors, Year, Country | Research Aim(s)/Research Question(s) | Sample Age No. of Participants Control/Experimental Groups–Yes/No? | Types of the Game(s) | Methods/Instruments | Major Findings | Pedagogical Implications | Limitations |
|---|---|---|---|---|---|---|---|
| **Rasti-Behbahani and Shahbazi, 2020 [67], Finland** | 1. Does a DGBVL task make a significant difference in the acquisition of aspects, dimensions, and scopes of the word-knowledge framework in comparison with a regular vocabulary learning task? 2. Which aspects, dimensions, and scopes of the word-knowledge framework are acquired significantly more efficiently? 3. To what extent are learned word-knowledge framework components correlated to one another? | 11–13 yr. 124 Yes | Role-playing commercial game | Quasi-experimental study (pre-test, treatment, and post-test), Oxford placement test | EG outperformed CG in vocabulary retention. The experimental group was more successful in acquiring productive recognition of form–meaning relationship, while the control group was more successful in acquiring receptive knowledge of association. | Implementation of DGBVL tasks into the teaching–learning process could be successful as they can enhance the acquisition of most components of the word-knowledge framework. | Participants might have learned some words from the earlier tests and that could have affected the answers. DGBVL could be used to develop other language skills and systems, as opposed to vocabulary. |
| **Ng and Raghbir, 2021 [62], Malaysia** | The study aims to investigate the Vocabulary Learning Strategies (VLS) used in a free-to-play massively multiplayer online role-playing game (MMORPG), Guild Wars 2, among Malaysian ESL players when they are playing and interacting with other players in the game's virtual world. | 23–24 yr. 4 No | MMORPG | Observation as participant, OBS software (Open Broadcaster Software) for analysis of observation | The most frequent vocabulary learning strategies utilized by the learners were: Consultation, incorporating words with Real-World, and using an online dictionary | Not available. | An insufficient number of participants for a representative sample of a big population (4). Only the genre of MMORPG has been covered. |

**Table 1.** *Cont.*

| Authors, Year, Country | Research Aim(s)/Research Question(s) | Sample Age No. of Participants Control/Experimental Groups–Yes/No? | Types of the Game(s) | Methods/Instruments | Major Findings | Pedagogical Implications | Limitations |
|---|---|---|---|---|---|---|---|
| **Fu et al., 2021 [61], China** | Explore what app features are preferred by non-English-major college students, and their perceptions about the effects of these apps and gamification settings on their vocabulary learning motivation and habits. | Major college students (age not disclosed) 55 No | Serious mobile games | Semi-structured interviews, content analysis, coding | Helpful to improve motivation and cultivate autonomous learning behaviours. Games increased self-confidence in language learning. Not all game elements have a positive effect on vocabulary learning—for example, competitiveness or overcomplexity. | The author suggests teachers add game elements to FLT based on the characteristics of different students to create a better learning experience and cultivate students' learning interests and autonomous learning behaviours. | Semi-structured interviews could be biased because they were based on students' self-reported perceptions. This study did not follow up with participants after a long period. The study only included participants from limited majors—need a bigger variety within research sample. |
| **Costuchen et al., 2022 [58], Spain** | To develop a serious game that would aid in the retention of foreign-language non-contextualized vocabulary items in different age categories. | 18–60 yr. 48 Yes | A serious game designed by the researchers | Quasi-experimental design (pre-test, treatment, and post-test), semi-structured interviews | EG outperformed CG in vocabulary post-test. Video games proved more efficient and entertaining in vocabulary learning than conventional methods. | A prototype—a serious game using the mnemonic system could offer an alternative to conventional study methods. | Incorporation of useful improvements due to time restraints. |
| **Dinçer and Dinçer, 2021 [59], Turkey** | Investigate the effect of a serious simulation game, X-Plane 11, offering an invaluable learning experience on aviation vocabulary acquisition. | 19–21 yr. 30 Yes | Serious game | Quasi-experimental design (pre-test, treatment, and post-test), semi-structured interviews | EG outperformed CG in vocabulary post-test. Video games proved more efficient and entertaining in vocabulary learning than conventional teaching methods. The serious game offers an efficient risk-free environment to broaden the learner's mental lexicon. | Implementation of simulator-based serious games could offer affordable and risk-free opportunities for learning, which could reduce anxiety in students and increase their learning achievement rate. | Number of participants. Counting in the vocabulary items of general English. Only the iPad version was included in the research. |

**Table 1.** *Cont.*

| Authors, Year, Country | Research Aim(s)/Research Question(s) | Sample Age No. of Participants Control/Experimental Groups–Yes/No? | Types of the Game(s) | Methods/Instruments | Major Findings | Pedagogical Implications | Limitations |
|---|---|---|---|---|---|---|---|
| **Calvo-Ferrer and Belda-Medina, 2021 [65], Spain** | Explore the effect of playing an online multiplayer social deduction game (i.e., a game in which players attempt to uncover each other's hidden role) on incidental and intentional second language (L2) vocabulary learning. | 16–18 yr. 54 Yes | Commercial social-deduction game | Quasi-experimental design (pre-test, treatment, and post-test) | An intentional approach to vocabulary learning may be more effective than incidentally encountering words during play. Advanced learners may benefit more from online multiplayer games with regard to learning new vocabulary. | Educators should provide learners with settings that promote engagement with new vocabulary and make students notice its form–meaning relationships. | Participants might have not been actively noticing the conversations that took place inside the game. The pre-test might have influenced the post-test results. Students were also supposed to offer the contextual meaning of the items within the vocabulary test, which may have not meant that they were completely ignorant of the words. Only completed on desktop PC, other platforms might have yielded different results. |
| **Octaberlina and Rofiki, 2021 [66], Indonesia** | Investigate the outcome of using an online game named SpellingCity to enrich the sample's vocabulary. | 17–20 yr. 22 No | Serious mobile game | Questionnaire, interview, pre-test, and post-test | DGBVL may be advantageous in vocabulary learning. Direction and instruction may increase the effectiveness of DGBVL even further. | Industrially created PC games can, with hypothetical direction, be adjusted for use in FLT process. Further support of EFL materials for the games could increase vocabulary retention. | Sample age—only tertiary students. |

**Table 1.** *Cont.*

| Authors, Year, Country | Research Aim(s)/Research Question(s) | Sample Age No. of Participants Control/Experimental Groups–Yes/No? | Types of the Game(s) | Methods/Instruments | Major Findings | Pedagogical Implications | Limitations |
|---|---|---|---|---|---|---|---|
| **Deris and Shukor, 2019** [64]**, Malaysia** | To investigate several existing mobile apps for language learning. Look into students' acceptance and the features of mobile apps conducive to vocabulary learning. | University students (age not disclosed) 3 interviewees and 30 questionnaire respondents No | Serious mobile games in general | Questionnaire, semi-structured interviews | Students have a positive acceptance of using serious mobile games in vocabulary learning. Learners sometimes feel overwhelmed by the vocabulary provided by serious mobile games; they find vocabulary too challenging. Internet connection is required to use the apps. | Game feature in mobile vocabulary learning apps provides an interesting and fun way of learning, while also motivating the students to learn more through mobile apps in the future. This could also increase the motivation of students to a learn language outside the classroom. | Number of participants Sample age—only tertiary students. Proficiency level—only A2 for the questionnaire. |

In this section, the four research questions set at the beginning of this review study are answered.

- **RQ1:** What are the main benefits and limitations of DGBVL?

In practice, several conclusions can be drawn from our systematic review. One of the main benefits of using digital games in education, in general, is the increase in motivation [24,47,57,61]. The gaming environment was identified to be engaging [45–48], creating mainly positive emotions [49], and creating autonomous language learning behaviours in students [61]. In this type of environment, students may feel less nervous about making mistakes [52], and it also has the ability to reduce foreign language anxiety [59,63]. Reduced anxiety in students has, in turn, proven to enhance memory and vocabulary retention [49,63,68]. Using digital games for vocabulary learning is in many cases used outside the classroom, and may build positive attitudes toward learning languages in general [60,62,64].

Within the disclosed data, studies subjected to this systematic review used a predominantly male sample. These studies mainly focused only on one age group of the sample, most often tertiary students. The number of participants was also often insufficient to draw broad generalizations about DGBVL. Furthermore, teachers need to be very careful when choosing the game to incorporate into the learning process, as it has to match the language proficiency of the target group, since the text coverage of the games may sometimes prove too difficult [68]. Another variable in incorporating DGBVL into the classroom is the choice of the right learning medium. Studies usually focus only on one learning medium. A comparison of the effectiveness of different digital media to use for DGBVL could make it easier for the instructors to decide which medium is the most appropriate for them. Additionally, researchers need to be careful when conducting the quasi-experiments about the sequence of the tests, as the research indicates that pre-tests may affect the post-test results [65,67].

- **RQ2:** What are the common trends researchers follow to conduct research in the area of DGBVL?

Most of the studies included in this systematic review followed a quasi-experimental design with an experimental and control group. The sampling methods most often chosen by the researchers were convenient sampling and purposive sampling. The sample age ranged from 11-year-old participants up to participants who were 60 years old. However, the most common sample included tertiary education students. This corresponds with the convenient sampling method as, arguably, for university instructors it may be the most accessible sample. In addition to the quasi-experiments, authors often used methods such as semi-structured interviews, questionnaires, and observations. This was mainly due to validation of the data and triangulation. As for the tools used, the results indicate that there are no standardized tools for testing vocabulary within the area of digital gaming.

- **RQ3:** Is DGBVL useful in learning English as a foreign language?

All of the studies conducted in the last 5 years and included in this systematic review reported that digital games had proven useful in vocabulary learning [57–59,65,67,68]. Additionally, all of the studies (where applicable) show that DGBVL proved even more effective than conventional teaching methods [57–59].

- **RQ4**: What are the pedagogical implications stemming from the available DGBVL research within the last 5 years?

The results reveal that games may reduce the anxiety students face while using the foreign language, may create a positive work environment, and may increase motivation. As many students continue using digital games outside the classroom, their overall vocabulary may be expanding even after the teaching/learning process is over. This opens doors to additional, extracurricular language learning. The effectiveness of DGBVL also highly depends on the correct choice of the particular game which is going to be used in the teaching/learning process. The game should match the proficiency level of students [68],

otherwise, they might have difficulties in understanding the text coverage. Calvo-Ferrer and Belda-Medina (2021) [65] suggest that instruction and making students notice the form–meaning relationships of the vocabulary items within the game can boost the effectiveness of DGBVL even further. Many scholars have been successful in designing a prototype of a vocabulary learning the digital game, which could be used within English lessons as an alternative to conventional teaching methods [56,58]. Furthermore, digital games can prove undemanding to be implemented into the educational process [68].

## 5. Discussion

Implementation of digital games in English language education may be an effective tool to liven up traditional English school lessons, while at the same time boosting their effectiveness. The use of digital games for learning vocabulary has proven more effective than using conventional teaching methods [57–59]. DGBVL proved to be an effective teaching method in the past [30,32–45], and was confirmed as such within this systematic review. Additionally, all of the studies (where applicable) show that DGBVL proved even more effective than conventional teaching methods [57–59]. However, this is in contrast to an older study carried out by Mayer (2014) [48], who suggested that serious games used in language learning did not surpass conventional teaching methods.

Within DGBVL research, gender has not been identified as a variable, which could affect the results of the research [65]. This contradicts an older study conducted by Rasti and Vahdat (2013) [41], which found that males were more successful in learning vocabulary through video games. Additionally, the results indicate that there are currently no standardized tools for testing vocabulary within the area of digital gaming. This might be due to the variable nature of digital games. A standardized vocabulary test for a specific game and language proficiency level would have to be designed.

Studies conducted by Yudintseva (2015) [69] and Poole and Clarke-Midura (2020) [70] reveal certain aspects that are similar to this review. Yudintseva (2015) [69] agrees that intentional learning strategies may further facilitate the method of instruction via digital games. These strategies may be even more effective if combined with interactions peers or natives. In their study, Yudintseva (2015) [69] focuses more on the incidental aspects of game-enhanced learning, rather than on the intentional strategies of English vocabulary instruction via the use of digital games (DGBVL). A section dedicated to pedagogical implications within our study provides a general approach towards advocating for the use of digital games in English language classrooms as a supportive material, or even as the main source of foreign language instruction.

In their systematic review, Poole and Clarke-Midura (2020) [70] analysed factors which are important to take under consideration while conducting research in the area of DGBVL. These factors include focusing on different languages, sample population size, focused research aspects, and effectiveness of the DGBVL approach. The findings of this systematic review contribute to the knowledge uncovered by the aforementioned study [70], by offering a summary of common research and methodological trends utilized within the DGBVL empirical studies.

The main limitation of this review study is that it covers a relatively short period of five years. However, as has been pointed out before, digital games in foreign language education became popular around 2018. Therefore, the authors limited the search to this period to collect only the most up-to-date knowledge. And yet, the findings generated conclusive information indicating that digital games, and thus DGBVL, are useful and should be incorporated in foreign language learning; as additionally, based on the findings of the detected studies, the results of this review illustrated that more research should be conducted in the area of methodological design.

## 6. Conclusions

In conclusion, the results show that digital games utilized in vocabulary learning have the capacity to increase motivation in students and thus engage them in vocabulary learning, both [24,47] inside and outside their foreign language classes. Additionally, they evoke mainly positive emotions in students and reduce their foreign language anxiety. This reduced anxiety has, in turn, been proven to enhance memory [50]. The effectiveness of DGBVL may surpass conventional teaching methods and may build autonomous learning behaviours in students [61].

There are several key takeaways, which could lead to further research in the area of DGBVL and to incorporation of this method into English learning classes in Europe, especially central Europe. Firstly, further studies should focus on identifying which serious or conventional digital games would prove to be most efficient in DGBVL, regarding learners' age and language proficiency. Consequently, standardized vocabulary tests for each game could be constructed to allow more objective vocabulary diagnoses within DGBVL. Secondly, focusing further research on the comparison of the effectiveness of different teaching–learning media in DGBVL could clarify which medium is optimal for this method and would be optimal for incorporation into the teaching–learning process. Additionally, more research is needed on the gender correlation within the outcomes of DGBVL. This additional research could possibly lead to the incorporation of DGBVL into the process of English instruction not as only a supplementary instruction method, but possibly as a full alternative for primary, secondary, and tertiary language education.

**Author Contributions:** Conceptualization: G.V. and B.K. Validation: G.V. and B.K. Formal analysis: G.V. and B.K. Investigation: G.V. Resources: G.V. and B.K. Data curation: G.V. and B.K. Writing—original draft: G.V. Writing—review and editing: G.V. and B.K. Visualization: G.V. and B.K. Supervision: B.K. All authors have read and agreed to the published version of the manuscript.

**Funding:** The study was supported by the SPEV project 2023.

**Institutional Review Board Statement:** Not applicable.

**Informed Consent Statement:** Not applicable.

**Data Availability Statement:** The data presented in this study are available in the results section.

**Acknowledgments:** The study was supported by the SPEV project 2023. The authors thank Veronika Doubravská for her help with data collection.

**Conflicts of Interest:** The authors declare no conflict of interest.

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
