# Peer review of "Exploring the Potential of Digital Game-Based Vocabulary Learning: A Systematic Review"

_systems, doi:10.3390/systems11020057_

Round 1
Reviewer 1 Report
Thank you for submitting your article “Exploring the Potential of Digital Game Based Vocabulary Learning: A Systematic Review” for review and consideration for publication in the Systems journal.
The present study conducted a systematic review to summarize the findings of all relevant individual studies on the application of digital games in teaching and learning vocabulary. I agree with the authors that digital games are very popular among all age groups as well as their well-established effectiveness in language education. With that in mind, it is apparent that many systematic reviews have been conducted in this domain, such as:
Vocabulary-specific paper: Game-enhanced second language vocabulary acquisition strategies: A systematic review (DOI: 10.4236/jss.2015.310015)
General-language paper: A Systematic Review of Digital Games in Second Language Learning Studies (DOI: 10.4018/IJGBL.2020070101)
Unfortunately, the authors failed to acknowledge the existing literature, which weakens the foundation of this manuscript. Thus, I recommend that the authors examine the existing systematic reviews and find a research gap or something that they have not addressed yet to warrant a new systematic review. The authors may contextualize the discussion deductively, e.g., from language education to vocabulary learning, or vice versa.
The statistical data cited from the Statistical Office of the European Union seems too long ago. Is there new data post-pandemic?
The authors discussed the concept and history of DGBL in the introduction. I recommend that they include examples of DGBL from early childhood to adult education to support their statement in the abstract: "digital games are very popular among all age groups". It would also give interesting insights for readers who may not be familiar with games as instructional technology. Some examples include:
Early Childhood Education: Kinder Learns: An Educational Visual Novel Game as Knowledge Enhancement Tool for Early Childhood Education (DOI: 10.18848/2327-0144/CGP/v27i01/13-34)
Adult Education: Utilizing Digital Educational Games to Enhance Adult Learning (DOI: 10.4018/978-1-5225-3132-6.ch009)
Both "game" and "play" have been used as keywords in searching documents. The authors should distinguish the difference between the two. Even the words "game" and "digital game" or "video game" entail different meanings in educational research.
Given that the Scopus database was a source of the review, why use ScienceDirect and not educational libraries such as Education Resources Information Center (ERIC)?
How did you assess the potential risk of bias in the 13 included studies?
Despite these revisions and questions that need addressing, the paper has merit. However, I am not entirely sure that it will be of interest to readers of the Systems journal. It may be more suitable for education journals (e.g., Education Sciences).
Author Response
Dear reviewer, Thank you so much for your insightful comments. We have done our best to implement all your suggestions and thus improved our manuscript. We included both studies in our manuscript, as well as identified the research gap between of both studies. Thank you once again for these constructive suggestions.
Please, find our responses within the attachment.

Reviewer 2 Report
This is an important review but the authors need to revise the review to address the following issues prior to publication:
1. The authors need to explain why the review period was set to be between 2018 and 2022. The identified 13 studies for the review are insufficient for a systematic review.
2. I am not sure if the review can answer the research questions 3 and 4. To answer research question 3, the authors need to conduct a meta analysis. Answers to Research Question 4 can be derived from the discussion of the review results but the question cannot be answered empirically by the review.
3. The writing in relation to the research questions in the discussion should be presented as results/answers to the research questions in the findings. The discussion should highlight on the implications of the review results for research and practice.
4. please
Author Response
Dear reviewer, Thank you so much for your insightful comments. We have done our best to implement all your suggestions and thus improved our manuscript.
Please, find our responses within the attachment.

Round 2
Reviewer 1 Report
I thank the reviewer for making a substantial revisions based on the given comments. I believe that the manuscript has improved significantly.
While all comments have been addressed, the point #3 was not totally answered. As recommended, citing sample digital games and dissecting why they are successful as a pedagogical tool would paint a common picture for the readers. Other than this, I believe the paper has merit for publication.
Author Response
Dear reviewer, thank you for your comment.
Based on your suggestions, we have done our best to improve the study by adding examples on how may digital games be useful in regard to different age groups (children and adults).
Please see the attachment to see the responses to your comments.

Reviewer 2 Report
Regarding the authors' response to Point 2: Point 2: The identified 13 studies for the review are insufficient for a systematic review.
I do not think that it is appropriate ot use their own work to prove that they are right.
Regarding Point 3: The rewording of Research Question 3: how does the revised Question 3 differ from the 'benefits' part of Question 1? I still insist that the results should be answers to the research questions.
Author Response
Dear reviewer, thank you for your comments.
Please see the attachment to view the responses to your suggestions.

Round 3
Reviewer 2 Report
Thanks the authors for their revision. I have no further issue with the study.